# Feasibility of Developing Audiovisual Material for Training Needs in a Vietnam Orphanage: A Mixed-Method Design

**DOI:** 10.3390/ijerph20043118

**Published:** 2023-02-10

**Authors:** Patricia Jovellar-Isiegas, Carolina Jiménez-Sánchez, Almudena Buesa-Estéllez, Pilar Gómez-Barreiro, Inés Alonso-Langa, Sandra Calvo, Marina Francín-Gallego

**Affiliations:** 1Department of Physical Therapy, Faculty of Health Sciences, Universidad San Jorge, Villanueva de Gállego, 50830 Zaragoza, Spain; 2Department of Physiatry and Nursing, Faculty of Health Sciences, IIS Aragon, University of Zaragoza, 50001 Zaragoza, Spain

**Keywords:** disabled children, international cooperation, online training, mixed-method research, physical therapy

## Abstract

Disabled children living in orphanages in low-income countries may not have access to the therapy they need. The COVID-19 pandemic has complicated the situation dramatically, making online training activities a possible innovative option to meet the real needs of local staff. This study aimed to detect the training needs of the local staff of an orphanage in Vietnam, as well as develop an audiovisual training material and measure its feasibility. Training needs were identified through a focus group carried out by the volunteers of Fisios Mundi, a nongovernmental organization. The audiovisual training material was developed to meet these specific needs. Lastly, its feasibility was evaluated, in terms of both content and format, through an ad hoc questionnaire. Nine volunteers participated in the project. Twenty-four videos were created and structured around five themes. This study expands the body of knowledge on how an international cooperation project can be developed in a pandemic situation. The audiovisual training material content and format created in this project was considered by the volunteers as very feasible and useful for training the staff of a Vietnamese orphanage.

## 1. Introduction

At present, approximately 34% of infant death takes place in South Asia. Vietnam is placed in 40th position with over 63,000 deaths per year [1]. Seven percent of Vietnam’s total population suffers from a disability and 13% of the total population cohabitates with a disabled person at home [2]. In 2003, most infant deaths were due to preventable causes such as neonatal disorders and pneumonia [3]. Despite the fact that these two main causes can be treated with physical therapy, the country has no, or very limited, access to physical therapy professionals or medical funds [4].

In the framework of international cooperation (IC), nongovernmental organizations (NGOs) are playing an increasingly important and dynamic role. IC projects imply all efforts made between two or more countries to address a problem [5]. IC in health development is still needed today to become conscious about children’s human rights in many developing countries [6]. NGOs are particularly committed to achieving vulnerable population coverage, mainly through their active participation in providing health services [7]. Orphanages and foster care are the options for children who have no family to live with. These institutions are often under-supported and underserved [8]; hence, NGOs react by providing prevention and intervention programs. Children, especially those living in orphanages, are often the highest priority, because there is limited capacity to take care of their basic needs such as feeding and clothing, as well as other needs such as therapeutic and physical treatments [9]. Therefore, the prevention and early treatment of childhood diseases in low-income countries is of vital importance as it will improve health conditions, both for the children and for their families [10,11]. In this way, most rehabilitation programs implemented in NGO projects, in the field of disability, focus on personal needs, making the involvement of all stakeholders (people with disabilities, caregivers, and their representative organizations) necessary [11].

Although there is not much scientific literature on the issue, some previous studies have identified needs presented by children living in orphanages. The study of Rubin et al. [12] assessed the overall dental treatment needs of children living in an orphanage in Uganda. Simşek et al. [13] examined the prevalence of emotional and behavioral problems and associated factors in children that were reared in orphanages. Additionally, Hermenau et al. [14] evaluated the mental health problems of children who lived in orphanages considering the need for specialized knowledge and clinical competence for assessment and treatment of these problems by clinicians [15]. Regarding rehabilitation projects implemented in IC, Garrido-Ardila et al. [11] conducted an online questionnaire for professionals in the field of physical therapy to learn about orthopedic treatment for children with cerebral palsy. 

However, this critical situation has been made more difficult by the COVID-19 pandemic due to the restrictions on travel and face-to-face contact, which prevent physical therapists from treating the children [16]. The global health crisis due to the COVID-19 pandemic has also affected the functioning of NGOs [17]. The impact of nonaccess to healthcare for children during the COVID-19 pandemic requires urgent collaboration and cooperation to provide adequate health coverage [18]. In addition, the COVID-19 pandemic increased the need for health-related vigilance in the community, which involves the extra responsibility of caring for children in need of protection [18,19].

In this way, since NGOs were already intimately connected with their constituents, they did their best to respond to immediate needs in new ways, exploring alternatives to face-to-face assistance [20]. Thus, the digital era has allowed us to continue online training activities with beneficial results, even in IC projects [21]. The World Confederation for Physical Therapy (WCPT) defined digital practice as “a term used to describe healthcare services, support, and information provided remotely through digital communication and devices” [22]. Thus, the COVID-19 pandemic served as an opportunity to find innovative ways to develop health promotion and community-based care. In this respect, the literature indicates that the use of digital technologies to deliver medical care, health education, and public health has been useful during emergencies and disasters, allowing for remote consultation, monitoring, and management [23,24,25]. Moreover, the use of distance learning has proven to be an effective and successful service under COVID-19 pandemic conditions [25]. Therefore, alternative approaches can be implemented without traveling to the destination country, making it possible for NGOs to continue collaborating with developing countries.

However, to the best of our knowledge, during the COVID-19 pandemic, no studies analyzed the training needs of local staff in orphanages, not receiving help from NGO professionals. Additionally, there are no studies that analyzed the possible training of these workers through audiovisual material created by NGOs. The above gaps in the literature motivated this research. The aims of this study were (a) to detect the training needs of the local physical therapy staff from a Vietnamese orphanage, (b) to develop an online audiovisual physical therapy training material to be implemented in an orphanage, and (c) to analyze the feasibility of the online audiovisual training material from a volunteer point of view.

## 2. Materials and Methods

### 2.1. Study Design

This was a mixed-method, sequential derivative exploratory study that collected quantitative data according to the cross-sectional design with an underlying qualitative level, e.g., the concurrent nested method [26] (Figure 1). 

For the qualitative design, the Standards for Reporting Qualitative Research (SRQR) [27] were consulted, and the methodology for collecting quantitative data agreed with the STROBE statement [28].

The study was approved by the Ethics Committee of the Universidad San Jorge (Code 07-20/21) and was carried out according to the Helsinki Declaration [29].

### 2.2. Setting and Population

The volunteers from the NGO Fisios Mundi who participated in the study were selected using non-probabilistic convenience sampling, and they were recruited in October 2020. Those volunteers who showed interest and signed the informed consent became part of the study. The main researcher contacted the president of the NGO Fisios Mundi via email to explain the details and objectives of the study. Fisios Mundi contacted all the health professionals who had voluntarily participated in an IC project in health development in an orphanage in Vietnam between 2015 and 2019. 

### 2.3. Procedure

The study comprised three different phases (Figure 1). A diagnosis of training needs was first undertaken to identify the themes to be addressed. Secondly, the online training material was created with the objective of covering the identified needs. Thirdly, the feasibility of the created material was assessed from the point of view of the volunteers of the NGO Fisios Mundi through an ad hoc survey.

#### 2.3.1. Identifying Needs and Collecting Data: Focus Group

A semi-structured interview was conducted through a focus group composed of the volunteers of the NGO Fisios Mundi. The aim was to identify the training needs of the local staff caring for the children in the Vietnamese orphanage. 

The research team designed a guide of open-ended questions that were used to encourage participants to describe their experiences and thoughts in a flexible way [30]. As a result, the emergence of new themes was accommodated. The interview was conducted by one of the researchers who has experience in qualitative research. It took place via the Microsoft 365 Teams online platform. When the intervention of the participants ceased to provide new information, the interview was concluded [31]. The entire interview was audio-recorded and transcribed verbatim for subsequent qualitative data analysis.

#### 2.3.2. Creating the Resources: Online Audiovisual Training Material

The material (Figure 2) was developed considering the training needs of local Vietnamese staff, which were identified by the participants through the focus group. The research team oversaw the making of the videos and their subsequent editing. To produce this audiovisual material, the main components of audiovisual creation were considered (script, production, filming, post-production, and viewing) [32], and we focused on meeting the needs previously stated. This material was created to be as simple and understandable as possible for the target population (workers in orphanages in Vietnam). When the training material was completed, all participants had access to it through a link to the Microsoft 365 One Drive platform.

#### 2.3.3. Assessing the Feasibility of the Online Audiovisual Training Material: Ad Hoc Survey

An ad hoc survey (Appendix A) was conducted to assess both the content and format of the training material. The survey was sent to each participant via the Microsoft 365 Forms platform, and a 10 day deadline was set for participants to view the training videos and then respond to the survey.

Three clearly differentiated sections formed part of the ad hoc survey. The first section consisted of nine questions aimed at collecting the sociodemographic data of the participants and their experience in IC projects. The second section consisted of eight questions related to both the content and the format of the audiovisual training material. These questions were elaborated according to a five-point Likert scale [33]: 1 = totally disagree; 2 = disagree; 3 = neither agree nor disagree; 4 = agree; 5 = totally agree. Lastly, the third section was composed of three open questions about the positive aspects they identified in the training material, the aspects they thought could be improved, and their general opinion. It was estimated that 20–30 min would be necessary to complete the survey.

#### 2.3.4. Data Analysis and Statistical Methods 

Qualitative thematic analysis [34] was used to process the data collected through the focus group interview and the third section of the ad hoc survey. Data triangulation strategies were carried out to avoid bias in the analysis. 

Quantitative analysis was used to analyze the feasibility of the audiovisual training material through the ad hoc survey. Data analysis was conducted using SPSS (v. 25, SPSS Inc., Chicago, IL, USA). For Section 1 and Section 2 of the survey, descriptive statistics were represented as means, standard deviations (SD), numbers (n), and percentages (%). 

#### 2.3.5. Methodological Rigor

To control the rigor and quality of the qualitative design, several procedures were followed [35], such as the inspection of the transcriptions, favoring an environment of trust for data extraction, maintaining contact with the volunteers [30], developing a bracketing process [36], and the benefits of using a mixed design, which establishes a process of triangulation of methods, because some results are supported by others. Additionally, triangulation and peer-checking of the results obtained were carried out by our research team.

In addition, the SRQR [27], which helps to ensure validity, was used to obtain and to analyze the data and results. SRQR is a checklist used to assess the quality of qualitative studies. It consists of 21 items (standards) that evaluate the different sections of the investigation. These items are the result of the synthesis of 25 previous references. In the present project, the SRQR was used as a guide so that, when analyzing the methodological quality of the qualitative part of this mixed study, the standards of the best possible quality could be met.

## 3. Results

### 3.1. Sociodemographical Data of the Participants

The sociodemographic data of the study participants were obtained from the first section of the ad hoc survey, as presented in Table 1. The study had a sample of nine participants, of which eight were female and one was male. The median age was 33.0 years old (SD = 3.23). The professions in which they were trained were physical therapy (n = 7), occupational therapy (n = 2), psychopedagogy (n = 2), teaching (n = 1), and nursing (n = 1); four of them had double degrees.

### 3.2. Needs Identified through the Focus Group 

The focus group with the NGO volunteers took place in January 2021 and lasted 90 min. During the conversation that took place via the Microsoft 365 Teams online platform, the various specific needs of the Vietnamese orphanage children were detected. When asked about the needs existing related to the creation of the material, the volunteers agreed on the following themes:

Theme 1: Stimulating children and how to create stimulating materials. When the volunteers arrived at the different orphanages, they found many children with motor and cognitive disabilities. Not all the children were well stimulated as they spent most of the day in cribs or beds (described by the volunteers as “cages”). They said that one of the most important needs is to get the children out of these “cages” and start stimulating them. They pointed out that it would be interesting to teach them how to create stimulation material through everyday resources and materials.

“*For these children who are very affected, do more basal things, sensory stimulation, which is super easy. It can be done with bottles of things, with stones, with feathers, with… well, teach how to do it and with a sponge in hand, accompany them, and let them sponge themselves over their bodies, I don’t know” *
*(Volunteer 1)*


For this purpose, it was decided to create give videos showing local workers (1) how to make slime to play with your hands, (2) how to make sensory bags to stimulate the body, (3) how to make sensory bottles to promote visual stimulation, and (4) how to create musical bottles for auditory stimulation.

Theme 2: Encouraging group activities. A great stimulation opportunity is the relationship with the environment and with the other children. Volunteers affirmed that, in addition to individual therapy, they need to occupy the hours of the day with group activities to give the children opportunities to discover the context and feel free. This time is essential and much more therapeutic.


*“We had a lot of influence on all the boys being taken out of the room (…) trying to use the porches; the youngest and most affected children were taken out, we did stimulation outside, got some boys together and did something” *

*(Volunteer 4)*


In this case, one video with game ideas was created through exteroceptive information; another one was elaborated with activities in which movement is favored, and a last one showed group games.

Theme 3: Improving positioning and transfers. The volunteers reported how, on many occasions, the workers were overwhelmed and did not know how to manage some situations, mobilizing the children as if they were moving a burlap bag if a child was older or had a more severe disability and, thus, avoiding the opportunity to participate in group activities or stimulation time.


*“Yes, it is true that we dedicate ourselves a lot to training on postural control, by ‘taking them out, please’, by laying them on their backs, using wedges, taking them to chairs, etc.” *

*(Volunteer 7)*


Therefore, videos were created aimed at teaching them how children can be positioned functionally in bed through different decubitus positions, how to transfer to get out of bed, or simply, how to avoid contractures, deformities, and ulcers.

Theme 4: Improving habits in ADLs. A very important idea that the volunteers detected was the need to change some habits of ADLs that were not being performed well and posed serious complications for the children. Mealtime positioning was not correct; many of them had complications such as bronchoaspiration, which could be avoided by small changes. The volunteers detected the need to teach the correct way to feed the children, e.g., how to stimulate the muscles of the mouth, face, and neck, how to position the child, and how to offer food.


*“If they changed their habits a little or modified the way they did it and there was more postural control, they could avoid some problems such as bronchoaspirations at mealtime or some deformities, ulcers, things like that” *

*(Volunteer 3)*


Regarding this theme, one video was created to explain how to feed the children properly, two videos showed how to dress and undress them, and another two videos explained the correct positioning to play different games in the sitting position.

Theme 5: Promoting and teaching respiratory physical therapy techniques. Volunteers agreed on the need to train orphanage workers in some basic respiratory physical therapy techniques. Unfortunately, in the orphanage, some of the children died due to respiratory complications.


*“They had a great need in the subject of respiratory physio because many children were dying of respiratory complications, obviously also due to lack of mobility and such (…)” *

*(Volunteer 2)*


In relation to this topic, which was emphasized, different videos were created: (1) assessment of chest breath sounds through auscultation; (2) manual evaluation to identify the location of secretions; (3) upper airway clearance; (4) drainage of the middle airway; (5) provoked coughs to eliminate secretions; (6) respiratory games to promote ventilation.

### 3.3. Online Audiovisual Training Material Created

The development of audiovisual training material for the training of local staff working in the Vietnamese orphanage was the result of the analysis of the needs expressed by the volunteers. It consisted of 24 videos that were structured around five themes: “1. how to create stimulation material”, “2. group and therapeutic activities and games”, “3. positions and transfers”, “4. ADLs”, and “5. respiratory physical therapy” (Figure 2).

To facilitate the understanding of the contents and to avoid possible problems due to language, each video was between 30 s and 2 min long and was recorded without audio. Subtitles in Vietnamese and English were added. 

### 3.4. Feasibility of the Online Audiovisual Training Material through the Ad Hoc Survey

To analyze the feasibility of the online audiovisual training material from the volunteers’ point of view, the quantitative results of the ad hoc survey are offered. In the ad hoc survey used, the scoring questions in Section 2 were structured in two different blocks: video content and video format. In addition, the questionnaire included a third section with open-ended questions, which were analyzed through a qualitative research methodology.

### 3.5. Video Content

The results of the first block of the ad hoc survey can be found in Table 2. When asked about whether the content shown in the videos was adequate, and whether it was understandable to both the physiotherapists of the orphanage and the caregivers, 33.3% of the sample scored 4 (agree), and 66. 7% scored 5 (totally agree).

Regarding whether these videos could be applicable in the orphanage, 56% scored 4 (agree) and 44% scored 5 (totally agree).

Lastly, on whether these videos could help improve the assistance children receive from physiotherapists, 67% scored 4 (agree), and 33% scored 5 (totally agree).

### 3.6. Video Format

The results of the second block of the ad hoc survey are shown in Table 3. Responding to the question about whether the duration of the videos was correct, it was found that 22% scored 4 (agree), and 78% scored 5 (totally agree).

Regarding whether it was easy to access the materials proposed in the videos, 22.2% scored 3 (neither agree nor disagree), 22.2% scored 4 (agree), and 55.6% scored 5 (totally agree).

Lastly, on whether the written language used was favorable to facilitate the understanding of the videos, 11.1% scored 4 (agree), and 89.9% scored 5 (totally agree).

### 3.7. Open Questions (Qualitative Analysis)

Within the third section of the survey, the qualitative data obtained in the open questions were analyzed. Two main themes were gleaned from the responses of the volunteers regarding their opinion about the videos, which were (1) strengths and limitations of the videos, and (2) usefulness for the future.

#### 3.7.1. Theme (1) Strengths and Limitations of Videos

This theme includes comments from the participants referring both to the strengths of the audiovisual training material and to the aspects that they considered could be improved. Starting with the strengths of the videos, several participants endorsed their correct length:


*“I like the length of the videos” *

*(Volunteer 4).*


They also talked about the fact that they were attractive and entertaining videos and that they do not involve too much effort to access in orphanages:


*“The simplicity of the videos I think will make it easier for them to see them” *

*(Volunteer 6).*


The organization followed when structuring the videos was also highly valued by the participants:


*“Being organized, they can see each other again depending on the needs” *

*(Volunteer 6).*


Some participants considered that the way in which the concepts were explained was very adequate, commenting that they were simple, understandable, and clear videos.


*“The proposed activity is very clearly exposed” *

*(Volunteer 5).*



*“I really liked the focus and clarity of the videos” *

*(Volunteer 4).*


The materials and resources that were used were discussed; it was stated on several occasions that they were materials that the staff of the orphanages had easy access to.


*“Simple and easy materials that can be found anywhere” *

*(Volunteer 1).*


All the participants showed their appreciation for the videos. Some spoke about the feelings that seeing them had produced in them, reminding them of their experiences in Vietnam:


*“Thank you for your work and for making me remember the experiences I had in Vietnam” *

*(Volunteer 2).*



*“Congratulations on a job so well done!” *

*(Volunteer 1)*


Regarding the comments on aspects that could be improved, the videos on a specific topic, that of respiratory physical therapy, were discussed. It was mentioned that the complexity of the techniques and the fact that these videos included more text than the rest, combined with the little or no training that the orphanage staff had on this subject, could make it difficult to understand and carry out:


*“The respiratory ones are the ones that I find a bit complex, and I don’t know if it’s going to cost them a bit to understand since they have little training in that subject” *

*(Volunteer 1).*


It was also commented that the examples of children used in the videos might not meet all the needs of all the children in the orphanages, because of their size level, disability, etc.:


*“The examples in the video are with a small doll and, therefore, applicable to smaller and more manageable children; perhaps they see it hardly applicable for older children” *

*(Volunteer 2).*


One of the participants spoke about the fact that the length of some of the videos had been inadequate:
*“Maybe I would lengthen some videos or add more examples, or vice versa, other videos, I would shorten them a bit” **(Volunteer 5).*Lastly, it was suggested that the videos could have been made with real children:


*“Using children with special needs to observe the technical difficulties that each person may present “*

*(Volunteer 8).*


#### 3.7.2. Theme (2) Usefulness for the Future

This theme groups the comments regarding the possible usefulness of the videos in the future, emphasizing both the quality of the content they showed and their use in orphanages to meet needs that, today, cannot be met through face-to-face interventions since it is not possible to travel to the country.

The usefulness of the videos for the staff of the orphanages in Vietnam was highly appreciated by all participants:


*“I think this material will be of great help to the professionals of the orphanage” *

*(Volunteer 5).*



*“I think they can be a very motivating tool” *

*(Volunteer 3).*



*“The simplicity of the videos I think will facilitate their application”*

*(Volunteer 6).*


It was said that the content met the needs that were discussed in the focus group:


*“I think that all the topics discussed in the group are addressed and you have found how to transmit it in a very pleasant and clear way”*

*(Volunteer 2).*


It was also commented that these videos could be used as a possible intervention in the orphanage until the pandemic allows once again people to travel to the country:
*“I believe that this material can contribute to give continuity to the project during the period in which it is not possible to travel to Vietnam”**(Volunteer 7).*
*“I think you will both refresh concepts that have already been worked on in our training sessions and you will provide them with new ideas and ways of working to continue learning and improving their treatment and treatment with children” **(Volunteer 5).*
*“I think it is a very viable and appropriate way to replace face-to-face training”**(Volunteer 7).*Lastly, it was considered that these videos could be useful not only in the orphanages of Vietnam but also in different areas around the world:

“Very valid not only for Vietnam but also for the rest of the world, at both the volunteer and the university level” 
*(Volunteer 8).*


## 4. Discussion

In 2020, the COVID-19 pandemic accentuated inequalities, particularly those affecting the most vulnerable populations and the NGOs that provided them with assistance. As a result, NGOs were unable to send professional volunteers who could train the orphanage workers and ensure the proper care of the children. Given the care gap, this study proposed to generate specialized online training for orphanage workers and to collaborate with Fisios Mundi NGO, jointly reaching the goal of ensuring the maintenance of adequate intervention in the orphanage.

The authors found other studies that also examined children’s needs in orphanages [12,13] or the need for specialized training for the staff who provide care for them [14] in the literature. However, this is the first study that analyzed the specialized training needs of the physical conditions of children during the COVID-19 pandemic and that offered an alternative solution—the creation of audiovisual materials to the workers.

In relation to the results of the focus group that analyzed the training needs of local staff from the point of view of the NGO volunteers, all the volunteers agreed on five themes: stimulating children and how to create stimulating materials; encouraging group activities; improving positioning and transfers; improving habits in daily life activities; promoting and teaching respiratory physical therapy techniques. 

### 4.1. Stimulating Children and How to Create Stimulating Materials

Participants emphasized the importance of stimulating children and proposed some simple possibilities *(“For these children who are very affected, do more basal things, sensory stimulation. It can be done with bottles of things, with stones, with feathers (…)”. Volunteer 1*). Furthermore, 43% of children under 5 years of age in low- and middle-income countries are at risk of not achieving their developmental potential. In this way, it should be highlighted that early learning and stimulation in the first 2 to 3 years of life play a key role in children’s development. Moreover, low-cost activities with household or handmade objects could promote early development of children as proposed by participants of our study [37,38]. 

### 4.2. Encouraging Group Activities

Although the target group in our study were children, it is known that the prevalence of psychological symptoms in adolescents living in orphanages is higher than in the general adolescent population. As demanded by the volunteers in the focus group, encouraging children or adolescents to participate in sports and in group activities, and improving sports facilities in orphanages are interventions that can promote mental health in both populations [39].

### 4.3. Improving Positioning and Transfers

Despite the broad geographic variation, research shows significant health problems in children living in orphanages (ulcers, sleeping disorders, etc.), many of which could be preventable and/or treatable [40]. For example, pressure ulcers remain common in the community. Moreover, pressure ulcers often become chronic injuries that are difficult to treat and that tend to recur after healing. Taking into account these challenges, the orphanage staff should have the knowledge and skills to treat them. This is in agreement to the needs identified in the focus group of this study, where the staff and volunteers demanded training to improve the children’s positioning and transfers to decrease ulcers and to improve conditions for children *(“Yes, it is true that we dedicate ourselves a lot to training on postural control (…)”. Volunteer 7).*

### 4.4. Improving Habits in ADLs

The results about habits in ADLs show how important is to modify some food-related aspects. The literature regarding the health screening of children in orphanages is limited and screening activities are infrequent. The pandemic aside, the study of Shanti et al. [41] evaluated the risk factors and treatment needs of orphan children of Selangor, Kuala Lumpur, Malaysia, identifying poor oral health. These results are similar to ours focus group, where one of the needs identified was the improvement of daily habits, e.g., the correct way to feed the children (“*If they changed their habits a little or modified the way, they could avoid some problems (…)”. Volunteer 3)*. A few studies describing health screening performed in orphanages in Eastern Europe, Ethiopia, Ghana, and Haiti demonstrated previously undiagnosed malnutrition, anemia, respiratory problems, hepatitis B and C, visual and hearing disorders, skin infections, and dental caries [42,43,44]. 

### 4.5. Promoting and Teaching Respiratory Physical Therapy Techniques

During 2012, two orphanages in Vietnam suffered an outbreak of severe acute respiratory infections [45]. Moreover, the increasing number of children adopted internationally provides a unique situation for the spread of emerging infections, indicating the precarious situation of the children living in orphanages [46]. This is consistent with volunteers’ experiences at the orphanage and the demand for respiratory physical therapy techniques that they indicated in the focus group *(“They had a great need in the subject of respiratory physio because many children were dying of respiratory complications (…)”. Volunteer 2)*.

In relation to the online audiovisual training materials, they were created to cover the needs identified, to help the staff to increase their knowledge, and to improve their skills in caring for the children. Due to the pandemic, the staff of the orphanage could not count on the help of volunteers who in other situations collaborated with them in the treatment of children; thus, a series of videos were created. To the authors’ knowledge, only the study of Hunt et al. previously developed audiovisual material through which several successive interventions were conducted in the upbringing of babies in an orphanage in Tehran, with beneficial effects versus home-reared American children from predominantly professional families [47].

### 4.6. Ad Hoc Survey

The ad hoc survey was created due to the need for a quantitative evaluation, and in the absence of a scale to assess the feasibility of the audiovisual training material used. The acquisition of new knowledge in the orphanage through the audiovisual training material created could be more favorable when it is organized by themes, which was confirmed by the ad hoc survey results. In relation to the quantitative results, they indicated a medium–high level of agreement and showed that the volunteers who evaluated the audiovisual training material found it to be an adequate material for the online training of the orphanage staff (“*I believe that this material can contribute to give continuity to the project during the period in which it is not possible to travel (…)”. Volunteer 7*). Thus, the use of videos is a trending tool in telerehabilitation. Minghelli et al. showed how, during the COVID-19 pandemic, most physical therapists interrupted their face-to-face practices; however, most of them adapted to monitor their patients from a distance by making explanatory videos and, synchronous video conference treatment, as in our study [48]. Additionally, the study by Friedman et al. reported how having easy access to online content is related to a greater expansion of knowledge and skills, and patient-specific teaching was found to be better than generalized teaching [35].

Regarding the qualitative results obtained in the open questions of the ad hoc survey, two themes emerged: strengths and limitations of videos and usefulness for the future. 

### 4.7. Strengths and Limitations of Videos

As limitations of the videos, the excessive length of some of them, as well as the excess of text and the fact that they were not made with real children, was mentioned.

### 4.8. Usefulness for the Future

The material was considered very useful for the future by the participants, not only in the Vietnamese orphanage, as one of the volunteers’ comments showed (*“Very valid not only for Vietnam but also for the rest of the world, both at the volunteer and university level (…)”. Volunteer 8)*, but also in other areas where their needs might require it. In addition, its usefulness was commented as it could replace in-person training. In this way, Meuser et al. also concluded that the use of technology is a viable option when in-person interactions are not possible in special circumstances [49]. This supports the hypothesis that was stated in this study—that the audiovisual training material created will be able to continue to help and train the staff of the orphanage, seeking to improve the care they offer to the children. 

Collectively, our results showed how it was possible to collect, in the videos created, all the real needs existing in the orphanage, transmitting the necessary knowledge to help the staff in a simple, pleasant, and clear way and, thus, fulfilling the main objectives of the study, as reflected by the volunteers in the focus group. Heijnen et al. showed how rehabilitation workers in developing countries encounter treatments that are not feasible in their countries, which can lead to frustration [50]. For this reason, it is necessary to develop training materials that contemplate the real needs of local personnel so that they can implement them in the context in which they find themselves, as in our study, where this type of material created during the pandemic allowed a benefit [49].

However, it should be noted that the results obtained in this study should be interpreted with caution and cannot be generalized, as they were confined to a single NGO and a single orphanage in Vietnam.

## 5. Conclusions

This study expands the body of knowledge on how an IC project can be developed in a pandemic situation, according to the detection of the real needs of local staff and creating audiovisual training materials considering the resources available in their real context. 

The audiovisual training material created on the basis of the needs detected in this project (“how to create stimulation material”, “group and therapeutic activities and games”, “positions and transfers”, “ADLs”, and “respiratory physical therapy”) was considered from the volunteers’ point of view as very feasible and useful, in terms of both content and format, for training the staff of a Vietnamese orphanage.

Future research should aim to implement this training and to measure the degree of satisfaction that local staff experience with this online approach.

## Figures and Tables

**Figure 1 ijerph-20-03118-f001:**
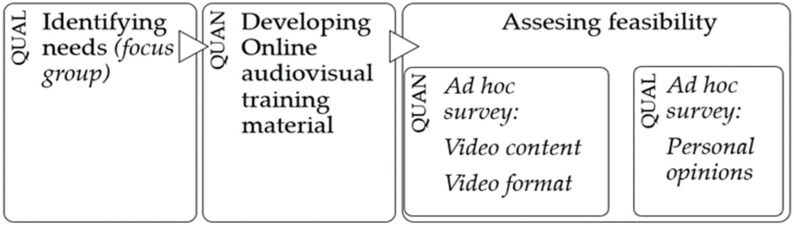
Study design. Methodology and study phases. QUAN = quantitative, QUAL = qualitative.

**Figure 2 ijerph-20-03118-f002:**
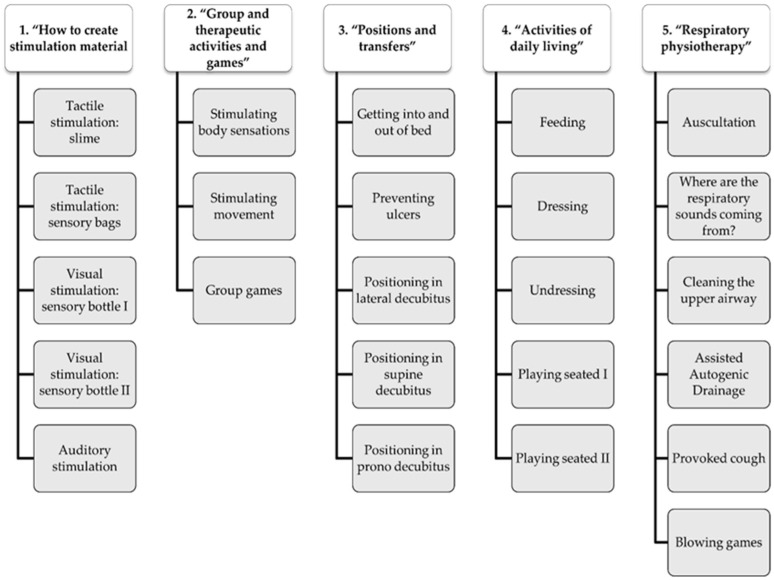
Audiovisual training material. Consisting of 24 videos distributed across the five themes based on the needs identified by the volunteers in the focus group.

**Table 1 ijerph-20-03118-t001:** Sociodemographic characteristics of the Fisios Mundi volunteers (n = 9).

Characteristics	Mean	SD
**Age**	33.0	3.23
**Gender**	**n**	**%**
Female	8	88.9
Male	1	11.1
**Profession**		
Physiotherapist	7	54.0
Occupational therapist	2	15.0
Psychopedagogist	2	15.0
Teacher	1	8.0
Nurse	1	8.0
**Years of working experience**		
Less than 2 years	0	0
2 to 4 years	0	0
5 to 7 years	2	22.2
More than 7 years	7	77.8
**Was Vietnam their first international cooperation experience?**		
Yes	0	0
No	9	100
**How long did their intervention in Vietnam last?**		
1 week	1	11.0
2 weeks	6	67.0
3 weeks	2	22.0
More than 3 weeks	0	0

**Table 2 ijerph-20-03118-t002:** First block of the ad hoc survey: video content. n: number; %: percentage; score: 4 = agree, 5 = totally agree.

Ad Hoc Survey Items	Score	n	%
The content shown (topics addressed) in the videos is appropriate and can contribute to meeting the real needs of the orphanage	4	3	33.3
	5	6	66.7
The content of the videos is understandable/comprehensible for the physiotherapists at the orphanage	4	3	33.3
	5	6	66.7
The content of the videos (especially that related to feeding, transfers and clothing) is understandable/comprehensible to the caregivers in the orphanage	4	3	33.3
	5	6	66.7
The content of the videos and the proposals made in them are applicable to the orphanage	4	5	56.0
	5	4	44.0
The use of these videos can contribute to improving the training and assistance that physiotherapists offer to the children in the orphanage	4	6	67.0
	5	3	33.0

**Table 3 ijerph-20-03118-t003:** Second block of the ad hoc survey: video format. n: number; %: percentage; score: 4 = agree, and 5 = totally agree.

Ad Hoc Survey Items	Score	n	%
The length of each video seems to be adequate	4	2	22.0%
	5	7	78.0%
The materials used in the videos are appropriate and can be used in the orphanage in Vietnam	3	2	22.2%
	4	2	22.2%
	5	5	55.6%
The written language used in the videos is appropriate and contributes to a better understanding of the content	4	1	11.1%
	5	8	89.9%

## Data Availability

Not applicable.

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
