# Peer review of "Feasibility of Developing Audiovisual Material for Training Needs in a Vietnam Orphanage: A Mixed-Method Design"

_ijerph, 2023, doi:10.3390/ijerph20043118_

Round 1

Reviewer 1 Report

Comments: 

in Section 3, methodology, what is the purpose of including Table 1? The table shows  extensive info about the demographic features, however, the only feature mentioned is the age and gender. so what is the purpose of the Table and the rest of the info? if not discussed remove, if important, pls state in the interpretation of the table. 

Results:

3.2. Needs identified through the focus group

...material through everyday resources and materials. For example, In Extract 1 below , volunteer 1 explains ....( state what the extract shows? what is the point being highlighted here?) 

"For these children who are very affected, do more basal things, sensory stimulation, which...

you need to link the interpretation with the extract , otherwise you are presuming that the reader would make sense of what the volunteer said and what point you are trying to make. 

pls revise the whole section accordingly 

a brief summary of the main points at the end of the section would be good, before starting section 3.3

Section 3.3

Figure 2 is given but again not discussed. you have stated the five themes, and that 24 videos were created, you do not however discuss the 24 videos. For example under 1. how to create simulation material, you have given 5 videos, but there is no mention of what they are, again assuming that the reader would interpret based on the figure, but that is not reader friendly, pls state what the figure is all about in more detail 

after each sub section, pls summarize the main points of the findings, before going  to the next subsection 

General comments:

All Tables and Figures need to be elaborated on. 

if Extracts from interviews are given, they must be labeled and mentioned in the interpretation of the findings 

all the subsections in the findings should include a statement or two as a summary of the main findings before going to the next subsection 

the paper still needs to be revised, and checked for spelling and grammar 

overall a much improved paper. 

Author Response

Detection of training needs and development of audiovisual training material from the perspective of NGO Fisios Mundi volunteers: a Service-Learning Project

We would like to thank the reviewers for reviewing our manuscript. We also appreciate the

reviewers for their constructive comments to improve the manuscript´s quality. We have carried out the changes that the reviewers requested and revised the manuscript accordingly.

Please find attached a point-by-point response to reviewer’s concerns. We hope that you find our responses satisfactory, and that the manuscript is now acceptable for publication.

Reviewer 1

  • Comments:

in Section 3, methodology, what is the purpose of including Table 1? The table shows  extensive info about the demographic features, however, the only feature mentioned is the age and gender. so what is the purpose of the Table and the rest of the info? if not discussed remove, if important, pls state in the interpretation of the table.

Response: The purpose of including Table 1 is to provide an overview of the descriptive characteristics of focus group participants who will later review the videos. We are interested in offering the reader notions about the professional experience of the participants as well as some information about their experience in the Vietnam cooperation project, to give a context to their narratives. Even so, to facilitate reading and considering your suggestion, we proceed to advance the paragraph that was found at the bottom of the table.

  • Results:
    • 2. Needs identified through the focus group.

...material through everyday resources and materials. For example, In Extract 1 below , volunteer 1 explains ....( state what the extract shows? what is the point being highlighted here?)

"For these children who are very affected, do more basal things, sensory stimulation, which... you need to link the interpretation with the extract , otherwise you are presuming that the reader would make sense of what the volunteer said and what point you are trying to make.

pls revise the whole section accordingly a brief summary of the main points at the end of the section would be good, before starting section 3.3

Response: In this case, for the presentation of results, the narrative that best reflects the content described in each of the topics has been chosen. It is not that there is only one narration per theme, but all those that were part of the same theme were categorized, reaching the description of the same, and the most representative of the descriptive paragraph previously written was selected.  That is why each chosen extract tries to reflect the views of the participants on each theme, which were used to develop the audiovisual training videos. Thanks to your questions, we have incorporated a paragraph below each theme where we hope that the link between the chosen extracts and the further development of the videos will now be clear.

  • Section 3.3

Figure 2 is given but again not discussed. you have stated the five themes, and that 24 videos were created, you do not however discuss the 24 videos. For example, under 1. how to create simulation material, you have given 5 videos, but there is no mention of what they are, again assuming that the reader would interpret based on the figure, but that is not reader friendly, pls state what the figure is all about in more detail after each sub section, pls summarize the main points of the findings, before going  to the next subsection.

Response: Thanks to your comment, the information has been expanded to emphasize each of the proposals included in the videos, relating to the needs detected by the volunteers.

  • General comments:

All Tables and Figures need to be elaborated on.

Response: We have revised the content so that the introductory paragraphs better situate the reader on the content of the figures and tables.

  • if Extracts from interviews are given, they must be labeled and mentioned in the interpretation of the findings all the subsections in the findings should include a statement or two as a summary of the main findings before going to the next subsection the paper still needs to be revised, and checked for spelling and grammar overall a much improved paper.

Response: Regarding to the reviewer’ suggestions, the discussion has been modified. Paragraphs have been moved to generate a better coherence in the text, and subsections with examples of the results have been created to provide that the discussion generated is based on the results obtained and is clear in the document.

Reviewer 2 Report

Slightly modify the methods and presentation of results.

Try to include more views of local workers

Keep in mind that it cannot be generalized, although the results are relevant due to the few studies that exist on how to improve stimulation and the health of children in orphanages.

Author Response

Detection of training needs and development of audiovisual training material from the perspective of NGO Fisios Mundi volunteers: a Service-Learning Project

We would like to thank the reviewers for reviewing our manuscript. We also appreciate the

reviewers for their constructive comments to improve the manuscript´s quality. We have carried out the changes that the reviewers requested and revised the manuscript accordingly.

Please find attached a point-by-point response to reviewer’s concerns. We hope that you find our responses satisfactory, and that the manuscript is now acceptable for publication.

Reviewer 2

  • Slightly modify the methods and presentation of results.

Response: Regarding the methodology, we have reinforced the written explanation prior to each of the tables and figures to better situate the reader.

To clarify the presentation of the results, more information has been added on the relationship between the themes obtained in the focus group and the creation of each of the videos that support the detected need.

  • Try to include more views of local workers.

Response:  As you indicate, we tried to include more views from other local workers at the orphanage, especially the caregivers, before submitting the manuscript. However, this was not possible as the information had not been received. To this day we still have not received it.

  • Keep in mind that it cannot be generalized, although the results are relevant due to the few studies that exist on how to improve stimulation and the health of children in orphanages.

Response: We have modified the conclusion so that it is less generic and better specifies the clinical implication in our study. In addition, considering your comment, we have added a sentence at the end of the discussion stating that the results of our study cannot be generalized and should be interpreted with caution.
